# Influence of Estimated Glomerular Filtration Rate on Clinical Outcomes in Patients with Acute Ischemic Stroke Not Receiving Reperfusion Therapies

**DOI:** 10.3390/jcm10204719

**Published:** 2021-10-14

**Authors:** Mi-Yeon Eun, Jin-Woo Park, Bang-Hoon Cho, Kyung-Hee Cho, Sungwook Yu

**Affiliations:** 1Department of Neurology, Kyungpook National University Chilgok Hospital, Daegu 41404, Korea; eunmiyn@gmail.com; 2Department of Neurology, Korea University Anam Hospital, Korea University College of Medicine, Seoul 02841, Korea; parkzinu@korea.ac.kr (J.-W.P.); fevernakchal@naver.com (B.-H.C.); kh.cho.neuro@gmail.com (K.-H.C.)

**Keywords:** estimated glomerular filtration rate, chronic kidney disease, ischemic stroke, clinical outcome

## Abstract

Background: We aimed to determine whether estimated glomerular filtration rate (eGFR) is an independent predictor of clinical outcomes in patients with acute ischemic stroke not treated with reperfusion therapy. Methods: A total of 1420 patients with acute ischemic stroke from a hospital-based stroke registry were included in this study. Patients managed with intravenous thrombolysis or endovascular reperfusion therapy were excluded. The included patients were categorized into five groups according to eGFR, as follows: ≥90, 60–89, 45–59, 30–44, and <30 mL/min/1.73 m^2^. The effects of eGFR on functional outcome at discharge, in-hospital mortality, neurologic deterioration, and hemorrhagic transformation were evaluated using logistic regression analyses. Results: In univariable logistic regression analysis, reduced eGFR was associated with poor functional outcome at discharge (*p* < 0.001) and in-hospital mortality (*p* = 0.001), but not with neurologic deterioration and hemorrhagic transformation. However, no significant associations were observed between eGFR and any clinical outcomes in multivariable analysis after adjusting for clinical and laboratory variables. Conclusions: Reduced eGFR was associated with poor functional outcomes at discharge and in-hospital mortality but was not an independent predictor of short-term clinical outcomes in patients with acute ischemic stroke who did not undergo reperfusion therapy.

## 1. Introduction

Chronic kidney disease (CKD) is an established risk factor for cardiovascular disease that substantially contributes to morbidity and mortality [1,2]. A meta-analysis study also reported that low estimated glomerular filtration rate (eGFR), a commonly used criterion for the diagnosis of CKD, is an independent risk factor for incident stroke [3]. However, the impact of eGFR on the clinical outcomes of acute ischemic stroke is controversial, despite the substantial number of studies on this topic [4,5,6]. In some studies, patients with acute ischemic stroke with comorbid CKD defined by reduced eGFR had a worse functional outcome and higher mortality rate [7], as well as a higher incidence of symptomatic hemorrhagic transformation than patients without CKD [8]. In contrast, other studies have reported that reduced eGFR was not related to functional outcome or mortality at discharge or 90 days following acute stroke, after adjusting for other vascular risk factors [6,9,10]. One of the reasons for the conflicting results is the direct impact of reperfusion therapy on clinical outcomes that may also be influenced by eGFR. Although some studies have shown that no significant association exists between eGFR and clinical outcomes after reperfusion treatment, many clinicians are still reluctant to perform reperfusion therapy in patients with low eGFR. In fact, some studies have presented concerns related to reperfusion therapy for ischemic stroke in patients with renal dysfunction [11,12].

To evaluate the sole influence of low eGFR on the clinical outcomes of acute ischemic stroke, it is necessary to consider the effect of reperfusion therapy. However, only a few studies that have assessed the effects of eGFR on the clinical outcomes of ischemic stroke described the status of reperfusion therapy. Therefore, in this study, we aimed to evaluate the effect of eGFR on clinical outcomes (functional outcome, in-hospital mortality, and hemorrhagic transformation) at discharge in patients with acute ischemic stroke who did not undergo reperfusion therapy.

## 2. Materials and Methods

This study was a retrospective observational study based on a single-center stroke registry. The data from the patients with acute ischemic stroke or transient ischemic attack were prospectively collected in the registry of Korea University Anam Hospital. We collected the data of patients aged ≥18 years with acute ischemic stroke between January 2008 and March 2013 from the stroke registry. We excluded patients with transient ischemic attack and those with acute ischemic stroke managed with reperfusion therapy, including intravenous thrombolysis or endovascular reperfusion therapy. Patients without data on eGFR and clinical outcomes during admission were also excluded. The study protocol was approved by the Institutional Review Board of Korea University Anam Hospital (IRB no. 2015AN0343).

Data on demographics and vascular risk factors were obtained from the stroke registry and through a review of the patients’ medical records. Ischemic stroke was defined as a rapidly evolving focal neurologic deficit lasting >24 h or leading to death, with no apparent cause other than a vascular origin or ischemic lesions relevant to temporary focal neurologic symptoms on diffusion-weighted magnetic resonance imaging (MRI). Hypertension was defined as a repeatedly elevated blood pressure (systolic blood pressure ≥ 140 mm Hg or diastolic blood pressure ≥ 90 mm Hg) or use of antihypertensive medications. Diabetes was defined as fasting blood glucose level > 126 mg/dL, hemoglobin A1c level ≥ 6.5%, or use of insulin or oral hypoglycemic agents. Hyperlipidemia was defined as total cholesterol level > 200 mg/dL, low-density lipoprotein cholesterol (LDL-C) level > 100 mg/dL, or use of cholesterol-lowering medications. Current smoking was defined as being a smoker at the time of the index stroke. Atrial fibrillation was diagnosed based on electrocardiographic or 24 h Holter monitoring findings. Previous myocardial infarction (MI) was assessed based on the patients’ medical history.

All patients underwent detailed clinical assessments (conducted by a neurologist), laboratory tests, and radiologic evaluations. The initial stroke severity was assessed using the National Institutes of Health Stroke Scale (NIHSS). Laboratory examinations were performed on admission, including complete blood cell count, blood chemistry tests, and high-sensitivity C-reactive protein (hs-CRP), and homocysteine measurements. The lipid profile, including total cholesterol, LDL-C, high-density lipoprotein cholesterol (HDL-C), and triglyceride levels, were analyzed after at least 8 h of fasting. Cardiac evaluation and brain imaging, including computed tomography (CT) or MRI, were also performed. Vascular status was assessed using CT angiography, magnetic resonance angiography, or cerebral angiography. Stroke subtypes were classified using the TOAST (Trial of Org 10,172 in Acute Stroke Treatment) criteria [13]. Patients were managed according to the guidelines of the Korean Stroke Society or the American Heart Association/American Stroke Association [14,15].

Glomerular filtration rate (GFR) was estimated based on the blood samples drawn at the time of admission (median duration from symptom onset to admission: 12 h 18 min, interquartile range [IQR]: 2 h 54 min to 42 h 34 min) using the CKD Epidemiology Collaboration (CKD-EPI) formula, as follows: eGFR = 141 × min(serum creatinine/κ,1)^α^ × max(serum creatinine/κ,1)^−1.209^ × 0.993^age^ × 1.018 [if female] × 1.159 [if black], in which κ is 0.7 for women and 0.9 for men, α is −0.329 for women and −0.411 for men, “min” indicates the minimum serum creatinine level/κ or 1, and “max” indicates the maximum serum creatinine level/κ or 1 [16]. The patients were categorized into five groups according to eGFR, corresponding to CKD staging, as follows: ≥90, 60–89, 45–59, 30–44, and <30 mL/min/1.73 m^2^ [17].

The primary outcome was poor functional outcome at discharge, defined as a modified Rankin scale score of 3–5 (disability) or 6 (death). The secondary outcomes included in-hospital mortality, neurologic deterioration, and hemorrhagic transformation. In-hospital mortality was defined as death from any cause during hospitalization. Neurologic deterioration was defined as an increase in the NIHSS score of ≥4 points compared to the baseline score within 7 days after stroke onset. Hemorrhagic transformation was assessed on follow-up brain CT or MRI during hospitalization and defined as any ischemia-related cerebral hemorrhage within or outside a recent infarcted area.

Data are presented as means and standard deviations or medians and interquartile ranges for continuous variables and as numbers and percentages for categorical variables. The normality of the distribution of variables was assessed using the Kolmogorov–Smirnov test. One-way analysis of variance or the Kruskal–Wallis test was used to compare continuous variables, and the chi-squared test was used for categorical variables. Univariable and multivariable logistic regression analyses were performed to determine the association between eGFR and clinical outcomes. Traditional cardiovascular risk factors, as well as clinical and laboratory variables with a *p*-value of <0.2 in the univariable logistic regression analysis, were further examined using multivariable analysis. Results are presented as odds ratios (ORs) and 95% confidence intervals (CIs). All statistical analyses were performed using SPSS 23.0 for Windows (IBM Corporation, Armonk, NY, USA), and statistical significance was set at *p* < 0.05.

## 3. Results

### 3.1. Patient Characteristics

During the study period, 2254 patients were consecutively registered in the Korea University Stroke Registry. We excluded 185 patients with transient ischemic attack, 126 patients managed with reperfusion therapy, 360 patients who were not evaluated for eGFR, and 163 patients without clinical outcome data. After the exclusions, a total of 1420 patients were finally included in this study. Table 1 shows the descriptive summary statistics of all included patients and per subgroup according to the eGFR category. The median age of the patients was 69 years (interquartile range 60–76 years), and 851 (59.9%) patients were men. A total of 260 (18.3%) patients had an eGFR of <60 mL/min/1.73 m^2^, whereas 33 (2.3%) had an eGFR of <30 mL/min/1.73 m^2^. Patients with low eGFR were more likely to be older, to be female, and to have more vascular risk factors (e.g., hypertension, diabetes, and previous MI). Laboratory data showed that white blood cell (WBC) count and the levels of hemoglobin, hs-CRP, HDL-C, homocysteine, and uric acid differed according to the eGFR category. The hemoglobin level decreased, whereas the hs-CRP, homocysteine, and uric acid levels tended to increase as eGFR decreased. WBC count and HDL-C level were highest in the group with eGFR ≥30 and <45 mL/min/1.73 m^2^. A significant difference in the severity of stroke was observed, with patients with low eGFR having a significantly higher NIHSS score at admission (*p* = 0.001) (Table 1).

### 3.2. Clinical Outcomes

The clinical outcomes across the eGFR categories are shown in Table 2. A total of 479 (33.7%) patients had poor functional outcomes at discharge, and 32 (2.3%) patients died during hospitalization. Neurologic deterioration was observed in 54 (3.8%) patients, and hemorrhagic transformation was noted in 107 (7.5%) patients.

Significant differences were observed in poor functional outcome at discharge (*p* < 0.001) and in-hospital mortality (*p* < 0.001) across the eGFR categories (Table 2). Patients with low eGFR tended to have an increased risk of poor functional outcomes at discharge and in-hospital mortality. The incidence of hemorrhagic transformation increased as eGFR decreased. Neurologic deterioration showed similar trends, except in the lowest eGFR group. However, these differences according to eGFR categories were not statistically significant (*p* = 0.505 for neurologic deterioration and *p* = 0.102 for hemorrhagic transformation).

### 3.3. Associations between eGFR and Clinical Outcomes

In the univariable analysis, eGFR was significantly associated with poor functional outcomes at discharge (*p* for trend < 0.001, Table 3). Among other clinical and laboratory variables, age, sex, current smoking, atrial fibrillation, previous MI, hemoglobin level, WBC count, hs-CRP level, homocysteine level, uric acid level, and initial NIHSS score were significantly associated with poor functional outcome at discharge. However, the association between eGFR and poor functional outcome at discharge was no longer significant after adjusting for other variables (*p* for trend = 0.328, Table 4). In multivariable analysis, age (OR 1.02, 95% CI 1.01–1.04), uric acid (OR 0.87, 95% CI 0.77–0.98), and initial NIHSS score (OR 1.61, 95% CI 1.51–1.71) were independently associated with poor functional outcome at discharge.

Low eGFR was associated with in-hospital mortality in univariable analysis (*p* for trend = 0.001, Table 3). The association was no longer significant after adjusting for other risk factors in multivariable analysis (*p* = 0.240, Table 4). However, OR showed an increasing trend as eGFR decreased. Among potential confounding factors of in-hospital mortality, only uric acid level (OR 1.36, 95% CI 1.03–1.79) and initial NIHSS score (OR 1.19, 95% CI 1.12–1.26) were significantly associated with in-hospital mortality. In the univariable analysis, low eGFR tended to show an association with hemorrhagic transformation (*p* = 0.115, Table 3). The multivariable analysis also showed no significant association between eGFR and hemorrhagic transformation (Table 4). Age (OR 1.03, 95% CI 1.00–1.06), atrial fibrillation (OR 3.85, 95% CI 2.38–6.21), previous MI (OR 3.35, 95% CI 1.59–7.08), and initial NIHSS score (OR 1.13, 95% CI 1.09–1.17) were associated with hemorrhagic transformation. No significant association was observed between eGFR and neurologic deterioration in univariable (*p* = 0.532) and multivariable analysis (*p* = 0.931).

## 4. Discussion

The present study showed that decreased eGFR tended to increase the risk of poor functional outcome at discharge, in-hospital mortality, and hemorrhagic transformation in patients with acute ischemic stroke who did not undergo intravenous thrombolysis or endovascular reperfusion therapy. However, low eGFR was not associated with short-term clinical outcomes after adjusting for other vascular risk factors.

In this study, poor clinical outcomes more frequently occurred in the group with low eGFR, which is consistent with the results of previous studies [18,19]. Patients with low eGFR more frequently have a history of hypertension, diabetes mellitus, atrial fibrillation, and MI, which may explain why the risk of poor clinical outcomes in patients with ischemic stroke increases as eGFR decreases. However, eGFR did not remain an independent risk factor for clinical outcomes after adjusting for potential confounders in this study.

Therefore, the influence of eGFR on stroke outcome may be attributed to traditional comorbid risk factors rather than the effect of eGFR per se. Ischemic stroke and CKD share common traditional risk factors, including hypertension, diabetes, and dyslipidemia [20], and the pathophysiologic mechanisms of ischemic stroke and renal dysfunction may overlap to some extent. In other words, reduced eGFR may reflect vascular burden and maybe a mere bystander of the poor outcomes of acute ischemic stroke.

In fact, the effect of CKD on clinical outcomes in patients with ischemic stroke may be more highly related to renal structural damage than impaired kidney function. Although GFR is widely accepted as an overall index of kidney function, proteinuria or albuminuria is directly representative of renal structural damage associated with increased glomerular permeability and the incomplete tubular reabsorption of protein. Several studies have suggested that proteinuria or albuminuria, but not low eGFR, is an independent predictor of functional outcome and mortality in patients with ischemic stroke [6,10,21,22]. Several studies using categorized eGFR groups did not show an eGFR-dependent response in functional outcome or mortality after ischemic stroke and revealed an association only in the very-low-eGFR category [23,24]. In contrast, graded increases in neurologic deterioration, in-hospital mortality, and poor functional outcome after ischemic stroke were observed with higher degrees of proteinuria [10]. These findings suggest that, in addition to eGFR measurement, proteinuria assessment would provide a more accurate prediction of clinical outcomes.

The limitation of eGFR as an “estimated” value used for assessing kidney function could have affected our results. In this study, we calculated GFR using the CKD-EPI equation, which is considered a more accurate index for assessing GFR and a better predictor of cardiovascular mortality than the Modification of Diet in Renal Disease (MDRD) Study equation [16,25]. However, although the CKD-EPI equation has less bias than the MDRD Study equation, its accuracy remains limited, especially at an eGFR of >60 mL/min/1.73 m^2^. Therefore, underestimation or overestimation of GFR can occur and subsequently lead to the misclassification of patients, which may result in inconsistent findings across different studies. The use of more accurate measures of kidney function, such as eGFR calculated with the creatinine-cystatin C equation, may be important in future studies [26].

The strength of our study is the relatively large sample size of consecutive patients with acute ischemic stroke, which allowed their categorization into five eGFR groups. Most previous studies evaluated the effect of eGFR based on a cutoff of 60 mL/min/1.73 m^2^, which may not be sufficient to show an eGFR-dependent response in the prediction of the clinical outcomes of stroke [21,22,27]. However, the categorization of patients into five eGFR groups corresponding to CKD staging in this study revealed that low eGFR was not independently associated with poor functional outcome at discharge and in-hospital mortality, suggesting that eGFR could not be used as a predictive biomarker for short-term clinical outcomes in patients with acute ischemic stroke.

Our study had some limitations. First, as this was a retrospective study based on a single hospital registry, an inherent bias may exist. Second, eGFR on admission may not reflect the exact renal function, because it could be affected by the index stroke event, which could potentially induce malnutrition, deconditioning, and dehydration [28,29]. The relatively small number of patients in the extremely-low-eGFR group (<30 mL/min/1.73 m^2^) may not accurately represent a high-risk group and might have decreased the outcome events and statistical power. The possible influence of patients with prior mRS status and the effect of stroke subtypes might also need to be considered. However, the number of patients in each subgroup with worsening clinical outcomes was not enough to perform sufficient statistical analysis. Therefore, further studies with larger sample sizes and including other biomarkers for renal structural damage are needed.

## 5. Conclusions

This study demonstrated that low eGFR tended to increase the risk of poor functional outcome at discharge, in-hospital mortality, and hemorrhagic transformation in patients with acute ischemic stroke who were not treated with intravenous thrombolysis or endovascular reperfusion therapy. However, eGFR was not an independent predictor of short-term clinical outcomes. These results suggest that patients with acute ischemic stroke and low eGFR could be carefully managed similar to patients with a normal eGFR. Further large-scale studies on the effect of renal dysfunction on the clinical outcomes of patients with acute ischemic stroke, considering the complex interaction between eGFR and other vascular risk factors, as well as the effect of reperfusion therapy, are needed.

## Figures and Tables

**Table 1 jcm-10-04719-t001:** Baseline characteristics of the study population.

	All Patients (*n* = 1420)	eGFR ≥ 90 mL/min/1.73 m^2^ (*n* = 429)	60 mL/min/1.73 m^2^ ≤ eGFR < 90 mL/min/1.73 m^2^ (*n* = 731)	45 mL/min/1.73 m^2^ ≤ eGFR < 60 mL/min/1.73 m^2^ (*n* = 164)	30 mL/min/1.73 m^2^ ≤ eGFR < 45 mL/min/1.73 m^2^ (*n* = 63)	eGFR < 30 mL/min/1.73 m^2^ (*n* = 33)	*p*-Value
Demographics							
Age (years)	69 (60.0–76.0)	59.0 (51.0–66.5)	71.0 (65.0–77.0)	76.0 (70.0–81.0)	78.0 (71.0–85.0)	75.0 (67.0–81.0)	<0.001
Male sex	851 (59.9)	278 (64.8)	435 (59.5)	93 (56.7)	30 (47.6)	15 (45.5)	0.018
Vascular risk factors							
Hypertension	996 (70.2)	252 (58.7)	532 (73.0)	129 (78.7)	54 (85.7)	29 (87.9)	<0.001
Diabetes mellitus	467 (33.0)	129 (30.1)	234 (32.2)	64 (39.0)	25 (39.7)	15 (45.5)	0.086
Hyperlipidemia	541 (38.2)	157 (36.7)	283 (38.9)	65 (39.6)	23 (36.5)	13 (39.4)	0.936
Current smoking	454 (32.2)	195 (45.7)	214 (29.6)	32 (19.5)	6 (9.5)	7 (21.2)	<0.001
Atrial fibrillation	231 (16.3)	37 (8.6)	133 (18.2)	38 (23.2)	18 (28.6)	5 (15.2)	<0.001
Previous MI	64 (4.5)	12 (2.8)	31 (4.2)	10 (6.1)	7 (11.1)	4 (12.1)	0.006
Laboratory data							
eGFR (mL/min/1.73 m^2^)	81.5 (65.4–91.9)	96.9 (92.9–102.2)	77.5 (70.1–84.5)	53.7 (49.2–56.7)	40.1 (37.0–42.7)	25.3 (13.7–28.9)	<0.001
Hemoglobin (g/dL)	13.8 (12.7–14.9)	14.4 (13.3–15.4)	13.9 (12.8–14.9)	13.1 (11.9–14.0)	12.7 (11.5–13.6)	11.6 (10.4–12.4)	<0.001
WBC (× 10^3^/μL)	7.5 (6.0–9.1)	7.5 (6.0–9.4)	7.3 (5.9–8.9)	7.2 (5.7–8.8)	8.2 (7.2–9.7)	7.6 (6.6–9.4)	0.002
FBS (mg/dL)	113.0 (95.0–143.0)	113.0 (95.0–147.0)	113.0 (92.3–141.0)	113.0 (95.0–138.0)	122.0 (99.3–156.8)	121.0 (104.0–185.3)	0.379
hs-CRP (mg/L)	1.84 (0.70–5.55)	1.63 (0.64–4.30)	1.77 (0.70–4.89)	2.57 (0.95–13.29)	4.80 (0.91–31.11)	4.65 (1.20–31.74)	<0.001
TC (mg/dL)	169.0 (144.0–197.0)	171.0 (147.0–198.0)	170.0 (144.0–198.0)	162.0 (141.0–197.0)	168.0 (133.5–195.0)	164.0 (134.8–187.0)	0.472
HDL-C (mg/dL)	42.0 (35.0–50.0)	42.0 (36.0–52.0)	42.0 (36.0–50.0)	40.0 (34.0–47.0)	44.0 (34.0–54.0)	32.5 (29.0–41.8)	<0.001
LDL-C (mg/dL)	92.0 (74.0–110.0)	92.0 (75.0–109.0)	92.0 (76.0–111.0)	91.0 (74.0–110.0)	86.0 (62.5–106.0)	83.0 (72.8–95.8)	0.181
Triglycerides (mg/dL)	113.0 (82.5–159.0)	114.0 (80.0–161.3)	109.0 (82.5–157.0)	121.0 (86.0–161.0)	127.0 (77.0–173.0)	127.0 (88.3–181.85)	0.380
Homocysteine (μmol/L)	10.8 (8.3–14.0)	9.3 (7.5–11.7)	10.9 (8.5–14.1)	13.1 (10.9–16.5)	13.8 (10.7–17.1)	18.9 (13.4–21.9)	<0.001
Uric acid (mg/dL)	4.8 (3.7–5.7)	4.2 (3.3–5.1)	4.7 (3.8–5.6)	5.3 (4.6–6.7)	6.2 (5.3–7.5)	6.7 (5.4–9.2)	<0.001
Initial NIHSS score	3.0 (1.0–5.0)	2.0 (1.0–4.0)	3.0 (1.0–5.0)	3.0 (1.0–7.0)	3.0 (2.0–7.0)	4.0 (2.0–12.0)	0.001
mRS score at discharge	2.0 (1.0–3.0)	2.0 (1.0–3.0)	2.0 (1.0–3.0)	2.0 (1.0–3.0)	2.0 (1.0–4.0)	3.0 (1.5–4.0)	<0.001
TOAST							<0.001
LAA	497 (35.8)	165 (39.1)	246 (34.5)	53 (32.9)	19 (32.2)	14 (42.4)	
CE	278 (20.0)	72 (17.1)	154 (21.6)	35 (21.7)	14 (23.7)	3 (9.1)	
SAO	348 (25.1)	115 (27.3)	189 (26.5)	31 (19.3)	7 (11.9)	6 (18.2)	
Others	21 (1.5)	13 (3.1)	4 (0.6)	2 (1.2)	1 (1.7)	1 (3.0)	
Undetermined	244 (17.6)	57 (13.5)	120 (16.8)	40 (24.8)	18 (30.5)	9 (27.3)	

Values are presented as median (interquartile range) or number (%). *p*-values were calculated using the Kruskal–Wallis test for continuous variables or the chi-squared test for categorical variables. MI, myocardial infarction; eGFR, estimated glomerular filtration rate; WBC, white blood cell; FBS, fasting blood sugar; hs-CRP, high-sensitivity C-reactive protein; TC, total cholesterol; HDL-C, high-density lipoprotein cholesterol; LDL-C, low-density lipoprotein cholesterol; NIHSS, National Institutes of Health Stroke Scale; mRS, modified Rankin scale; TOAST, Trial of Org 10,172 in Acute Stroke Treatment; LAA, large-artery atherosclerosis; CE, cardioembolism; SAO, small-artery occlusion.

**Table 2 jcm-10-04719-t002:** Clinical outcomes in patients with acute ischemic stroke according to eGFR categories.

	All Patients (*n* = 1420)	eGFR ≥ 90 mL/min/1.73 m^2^ (*n* = 429)	60 mL/min/1.73 m^2^ ≤ eGFR < 90 mL/min/1.73 m^2^ (*n* = 731)	45 mL/min/1.73 m^2^ ≤ eGFR < 60 mL/min/1.73 m^2^ (*n* = 164)	30 mL/min/1.73 m^2^ ≤ eGFR < 45 mL/min/1.73 m^2^ (*n* = 63)	eGFR < 30 mL/min/1.73 m^2^ (*n* = 33)	*p*-Value
Poor functional outcome at discharge	479 (33.7)	120 (28.0)	243 (33.2)	67 (40.9)	28 (44.4)	21 (63.6)	<0.001
In-hospital mortality	32 (2.3)	5 (1.2)	11 (1.5)	9 (5.5)	5 (7.9)	2 (6.1)	<0.001
Neurologic deterioration	54 (3.8)	15 (3.5)	27 (3.6)	7 (4.3)	5 (7.9)	1 (3.0)	0.505
Hemorrhagic transformation	107 (7.5)	24 (5.6)	56 (7.7)	14 (8.5)	8 (12.7)	5 (15.2)	0.102

Values are presented as numbers (%). *p*-Values were calculated using the Kruskal–Wallis test. eGFR, estimated glomerular filtration rate.

**Table 3 jcm-10-04719-t003:** Univariable logistic regression analysis for clinical outcomes in patients with acute ischemic stroke.

	Poor Functional Outcome at Discharge		In-Hospital Mortality		Hemorrhagic Transformation		Neurologic Deterioration	
	OR (95% CI)	*p*-Value	OR (95% CI)	*p*-Value	OR (95% CI)	*p*-Value	OR (95% CI)	*p*-Value
Demographics								
Age (years)	1.04 (1.03–1.06)	<0.001	1.08 (1.04–1.12)	<0.001	1.05 (1.03–1.07)	<0.001	1.02 (0.99–1.04)	0.167
Male sex	0.62 (0.49–0.77)	<0.001	0.51 (0.25–1.04)	0.063	0.63 (0.43–0.94)	0.023	0.564 (0.33–0.97)	0.039
Vascular risk factors								
Hypertension	1.11 (0.87–1.41)	0.421	1.53 (0.66–3.56)	0.327	0.86 (0.57–1.31)	0.488	1.01 (0.56–1.83)	0.983
Diabetes mellitus	1.08 (0.86–1.36)	0.522	1.23 (0.59–2.53)	0.583	0.96 (0.63–1.46)	0.837	0.93 (0.52–1.67)	0.811
Hyperlipidemia	0.80 (0.63–1.00)	0.053	1.11 (0.54–2.26)	0.778	0.94 (0.62–1.41)	0.751	1.31 (0.76–2.56)	0.339
Current smoking	0.75 (0.59–0.95)	0.017	0.30 (0.10–0.85)	0.023	0.73 (0.47–1.14)	0.168	0.97 (0.54–1.74)	0.991
Atrial fibrillation	1.74 (1.31–2.32)	<0.001	4.79 (2.36–9.73)	<0.001	5.99 (3.97–9.03)	<0.001	2.25 (1.24–4.12)	0.008
Previous MI	1.67 (1.01–2.76)	0.047	2.25 (0.67–7.59)	0.191	3.06 (1.58–5.94)	0.001	1.26 (0.38–4.15)	0.706
Laboratory data								
Hemoglobin (g/dL)	0.87 (0.82–0.92)	<0.001	0.87 (0.73–1.03)	0.104	0.91 (0.82–1.01)	0.062	0.93 (0.81–1.07)	0.294
WBC (× 10^3^/μL)	1.12 (1.07–1.16)	<0.001	1.13 (1.04–1.23)	0.006	1.13 (1.06–1.19)	<0.001	1.07 (0.99–1.16)	0.112
FBS (mg/dL)	1.00 (1.00–1.00)	0.111	1.00 (1.00–1.01)	0.023	1.00 (1.00–1.00)	0.612	1.00 (1.0–1.01)	0.287
hs-CRP (mg/L)	1.02 (1.02–1.03)	<0.001	1.01 (1.01–1.02)	<0.001	1.01 (1.01–1.01)	<0.001	1.01 (1.01–1.02)	<0.001
Homocysteine (μmol/L)	1.02 (1.01–1.04)	0.005	1.02 (0.99–1.06)	0.203	1.01 (0.99–1.04)	0.269	0.99 (0.94–1.04)	0.574
Uric acid (mg/dL)	0.92 (0.85–0.99)	0.019	1.29 (1.06–1.57)	0.012	0.93 (0.81–1.07)	0.3	1.02 (0.86–1.22)	0.787
Initial NIHSS score	1.57 (1.49–1.66)	<0.001	1.22 (1.17–1.28)	<0.001	1.17 (1.14–1.20)	<0.001	1.1 (1.06–1.14)	<0.001
eGFR categories (mL/min/1.73 m^2^)		<0.001		0.001		0.115		0.532
eGFR ≥ 90	1		1		1		1	
60 ≤ eGFR < 90	1.28 (0.99–1.67)	0.062	1.30 (0.45–3.75)	0.633	1.40 (0.85–2.29)	0.182	1.02 (0.53–1.94)	0.957
45 ≤ eGFR < 60	1.78 (1.22–2.59)	0.003	4.92 (1.63–14.92)	0.005	1.58 (0.79–3.13)	0.194	1.23 (0.49–3.08)	0.657
30 ≤ eGFR < 45	2.06 (1.20–3.53)	0.009	7.31 (2.05–26.02)	0.002	2.46 (1.05–5.73)	0.038	2.38 (0.83–6.79)	0.105
eGFR < 30	4.51 (2.15–9.44)	<0.001	5.47 (1.02–29.35)	0.047	3.01 (1.07–8.50)	0.037	0.86 (0.11–6.74)	0.888

OR, odds ratio; CI, confidence interval; MI, myocardial infarction; eGFR, estimated glomerular filtration rate; WBC, white blood cell; FBS, fasting blood sugar; hs-CRP, high-sensitivity C-reactive protein; NIHSS, National Institutes of Health Stroke Scale; mRS, modified Rankin scale.

**Table 4 jcm-10-04719-t004:** Multivariable logistic regression analysis of the association between eGFR and clinical outcomes in patients with acute ischemic stroke.

	Poor Functional Outcome at Discharge		In-Hospital Mortality		Hemorrhagic Transformation		Neurologic Deterioration	
	Adjusted OR (95% CI)	*p*-Value	Adjusted OR (95% CI)	*p*-Value	Adjusted OR (95% CI)	*p*-Value	Adjusted OR (95% CI)	*p*-Value
eGFR (mL/min/1.73 m^2^)		0.328		0.240		0.582		0.931
eGFR ≥ 90	1		1		1		1	
60 ≤ eGFR < 90	0.94 (0.62–1.43)	0.784	0.31 (0.08–1.23)	0.095	0.73 (0.40–1.33)	0.305	0.90 (0.43–1.90)	0.788
45 ≤ eGFR < 60	1.01 (0.54–1.88)	0.987	0.92 (0.20–4.36)	0.920	0.52 (0.22–1.22)	0.130	0.94 (0.33–2.66)	0.900
30 ≤ eGFR < 45	1.23 (0.51–2.95)	0.649	0.46 (0.07–3.13)	0.428	0.49 (0.16–1.48)	0.207	1.20 (0.35–4.17)	0.772
eGFR < 30	3.00 (0.96–9.38)	0.059	0.23 (0.02–3.26)	0.275	0.76 (0.21–2.83)	0.683	0.43 (0.05–4.05)	0.459

Multivariable models were adjusted for variables with *p* < 0.2 in univariable analysis. eGFR, estimated glomerular filtration rate; OR, odds ratio; CI, confidence interval.

## Data Availability

The data presented in this study are available upon request from the corresponding author. The data are not publicly available, because of restrictions such as privacy or ethical issues.

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
