# Peer review of "Influence of Estimated Glomerular Filtration Rate on Clinical Outcomes in Patients with Acute Ischemic Stroke Not Receiving Reperfusion Therapies"

_jcm, 2021, doi:10.3390/jcm10204719_

Round 1
Reviewer 1 Report
In this article, Mi-Yeon E et al. studied the association between eGFR and clinical outcomes on patients with acute ischemic stroke not treated with reperfusion therapies. The subject is interesting, the Methods are appropriate, the Results are clearly presented, the Discussion is well structured and the article is globally well written. However, I would like to address some comments to the authors and suggest some changes before acceptance for publication.
1. Although the title of the article seems appropriate, "...patients with acute ischemic stroke without reperfusion therapy" sounds weird. I would suggest to revise the title. Maybe "...patients with acute ischemic stroke not treated with reperfusion therapy" or "...patients with acute ischemic stroke not receiving reperfusion therapies" would be better options.
2. The Methods are generally well described and collected variables are globally well defined. However I have some concerns in this section:
- I miss a statement about the type of study at the beginning: Though the database is prospectively collected, this is an observational retrospective study. I think it is important to state that at the beginning.
- eGFR is well defined as the principal exposure variable of the study. However, I miss specifying when it is measured (the time-point). Was it measured at admission in the emergency department (acute phase)? The first day of hospitalization? During the first week after the stroke? Before the stroke (considering medical reports)? This is important to specify.
- Please change "The main outcome..." for "The primary outcome..." and "The other outcomes..." for "Secondary outcomes include..."
- The primary outcome is mRS dichotomized at discharge (unfavourable when 3-6). Did you exclude patients with prior mRS >2? If not, your findings may be influenced negatively by including patients with prior mRS >2. Moreover, you are including patients who did not receive reperfusion therapies and having a prior mRS >2 can be an exclusion criteria for reperfusion therapies. So, I think it might had been a selection bias in your study population. You should discuss that in the Discussion section and if you have this information (prior mRS) I would strongly recommend you to perform a sensitivity analysis including only patients with a prior mRS 0-2. That would give quality to your work.
- Although it is frequent in the literature to equally use the term Multivariate and Multivariable for logistic regression analyses, in fact you are performing a Multivariable analysis (not a Multivariate). Please change it in Methods and Results
- You state: "Clinical and laboratory variables with a p-value of < 0.2 in univariate logistic regression analysis were further examined using multivariate analysis." How did you introduced the variables in the final model? Forcing them? Performing a stepwise selection modeling? Please specify.
- Please describe how did you managed confusion in the multivariable regression analysis (if you did).
Congratulations to the authors for this work.
Author Response
Q: Although the title of the article seems appropriate, "...patients with acute ischemic stroke without reperfusion therapy" sounds weird. I would suggest to revise the title. Maybe "...patients with acute ischemic stroke not treated with reperfusion therapy" or "...patients with acute ischemic stroke not receiving reperfusion therapies" would be better options.
A: We appreciate your suggestion. We changed the title per your recommendation: “Influence of estimated glomerular filtration rate on clinical outcomes in patients with acute ischemic stroke not receiving reperfusion therapies”.
Q: I miss a statement about the type of study at the beginning: Though the database is prospectively collected, this is an observational retrospective study. I think it is important to state that at the beginning.
A: We added the statement in the Methods section: “This study was a retrospective observational study based on a single-center stroke registry. The data from the patients with acute ischemic stroke or transient ischemic attack were prospectively collected in the registry of Korea University Anam Hospital (page 2, lines 58-60).”
Q: eGFR is well defined as the principal exposure variable of the study. However, I miss specifying when it is measured (the time-point). Was it measured at admission in the emergency department (acute phase)? The first day of hospitalization? During the first week after the stroke? Before the stroke (considering medical reports)? This is important to specify.
A: We agree with your concerns. We specified it as follows
: “Glomerular filtration rate (GFR) was estimated based on the blood samples drawn at the time of admission (median duration from symptom onset to admission: 12 h 18 min, interquartile range [IQR]: 2 h 54 min to 42 h 18 34 min (page 3, lines 96-98).”
Q: Please change "The main outcome..." for "The primary outcome..." and "The other outcomes..." for "Secondary outcomes include...".
A: We corrected these in the manuscript as you suggested: “The primary outcome was the poor functional outcome at discharge, defined as a modified Rankin scale score of 3-5 (disability) or 6 (death). The secondary outcomes include in-hospital mortality, neurologic deterioration, and hemorrhagic transformation (page 3, lines 105-107).”
Q: The primary outcome is mRS dichotomized at discharge (unfavorable when 3-6). Did you exclude patients with prior mRS >2? If not, your findings may be influenced negatively by including patients with prior mRS >2. Moreover, you are including patients who did not receive reperfusion therapies and having a prior mRS >2 can be an exclusion criteria for reperfusion therapies. So, I think it might had been a selection bias in your study population. You should discuss that in the Discussion section and if you have this information (prior mRS) I would strongly recommend you to perform a sensitivity analysis including only patients with a prior mRS 0-2. That would give quality to your work.
A: We agree with your opinion. Unfortunately, our data do not include prior mRS. Therefore, we were not able to analyze the effect of prior mRS scores on outcomes. We discussed this as a limitation in the Discussion section: “The possible influence of patients with prior mRS status and the effect of stroke subtypes might need to be also considered. However, the number of patients in each subgroup with worsening clinical outcomes was not enough to perform sufficient statistical analysis (page 9, lines 262-265). ”
Q: Although it is frequent in the literature to equally use the term Multivariate and Multivariable for logistic regression analyses, in fact you are performing a Multivariable analysis (not a Multivariate). Please change it in Methods and Results.
A: We corrected them. Thank you.
Q: You state: "Clinical and laboratory variables with a p-value of < 0.2 in univariate logistic regression analysis were further examined using multivariate analysis." How did you introduced the variables in the final model? Forcing them? Performing a stepwise selection modeling? Please specify.
A: To adjust all potential variables, we forcibly introduced traditional cardiovascular risk factors and other potential variables with a p-value of < 0.2 in univariable analysis. We added this information in the manuscript: “Traditional cardiovascular risk factors, as well as clinical and laboratory variables with a p-value of < 0.2 in the univariable logistic regression analysis, were further examined using multivariable analysis (page 3, lines 120-121).”
Q: Please describe how did you managed confusion in the multivariable regression analysis (if you did).
A: To manage the confusion, we used the multivariable regression analysis with all possible confounders and presented an adjusted odds ratio. We added this information throughout the manuscript. Thank you.

Reviewer 2 Report
The authors investigated whether low eGFR was associated with poor functional outcome at discharge, hemorrhagic transformation, or in-hospital mortality. However, in multivariate logistic regression analysis, they did not find reduced eGFR was not an independent risk of them above.
The manuscript does not have very high impact, therefore the authors have to reconsider the data and improve the manuscript.
- I am interested in the effect of reduced eGFR to each stroke subtype (e.g., LAA, CE, SAO, or intracerebral hemorrhage).
- I would say that increase in NIHSS score of >2 points would be defined as neurologic deterioration.
Author Response
We appreciate your valuable comments. This study was to evaluate the effect of eGFR on ischemic stroke patients, in particular, among the patients who did not undergo reperfusion therapy.
Q: I am interested in the effect of reduced eGFR to each stroke subtype (e.g., LAA, CE, SAO, or intracerebral hemorrhage).
A: We agree that the sub-analysis based on the TOAST classification would be important to further explore the effect of eGFR. However, the sample size for each group was not enough to perform an adequate statistical analysis. We were only able to analyze the effect of reduced eGFR for LAA and CE subtypes. The results showed that the eGFR did not remain a significant variable in each LAA and CE subtypes as our main results. The results of the sub-analysis are shown in the tables listed below (1-3).
1) Clinical outcomes based on the TOAST classification.
|
Classification (total subjects, n) |
Poor functional outcome at discharge |
In-hospital mortality |
Hemorrhagic transformation |
Neurological deterioration |
|
LAA (n=497) |
204 |
8 |
29 |
26 |
|
CE (n=278) |
104 |
14 |
56 |
16 |
|
SAO (n=348) |
60 |
0 |
2 |
1 |
|
Others (n=21) |
7 |
0 |
3 |
0 |
|
Undetermined (n=244) |
93 |
10 |
17 |
9 |
2) Univariable logistic regression analysis for clinical outcomes (Poor functional outcome at discharge) in patients with LAA and CE subtypes.
|
|
LAA |
CE |
||
|
OR (95% CI) |
p-value |
OR (95% CI) |
p-value |
|
|
Demographics |
||||
|
Age (years) |
1.04 (1.02-1.06) |
< 0.001 |
1.05 (1.02-1.07) |
< 0.001 |
|
Male sex |
0.75 (0.52-1.08) |
0.118 |
0.47 (0.28-0.76) |
0.002 |
|
Vascular risk factors |
|
|
||
|
Hypertension |
0.9 (0.61-1.34) |
0.612 |
1.13 (0.66-1.94) |
0.65 |
|
Diabetes mellitus |
1.16 (0.80-1.68) |
0.425 |
1.14 (0.65-1.99) |
0.65 |
|
Hyperlipidemia |
0.70 (0. 48-1.00) |
0.053 |
0.94 (0.55-1.59) |
0.806 |
|
Current smoking |
0.86 (0.59-1.25) |
0.419 |
0.69 (0.39-1.25) |
0.220 |
|
Atrial fibrillation |
1.44 (0.09-23.13) |
0.798 |
1.39 (0.83-2.31) |
0.210 |
|
Previous MI |
3.26 (1.22-8.71) |
0.019 |
0.58 (0.20-1.65) |
0.306 |
|
Laboratory data |
|
|
||
|
Hemoglobin (g/dL) |
0.91 (0.82-0.99) |
0.046 |
0.93 (0.81-1.07) |
0.303 |
|
WBC (× 103/μL) |
1.11 (1.04-1.18) |
0.002 |
1.23 (1.21-1.36) |
< 0.001 |
|
FBS (mg/dL) |
1.0003 (0.998-1.003) |
0.842 |
1.01 (1.004-1.018) |
0.001 |
|
hs-CRP (mg/L) |
1.02 (1.007-1.027) |
< 0.001 |
1.03 (1.01-1.04) |
< 0.001 |
|
Homocysteine (μmol/L) |
1.04 (1.01-1.07) |
0.01 |
1.05 (1.0-1.1) |
0.071 |
|
Uric acid (mg/dL) |
0.93 (0.83-1.05) |
0.259 |
0.89 (0.76-1.04) |
0.138 |
|
Initial NIHSS score |
1.65 (1.49-1.82) |
< 0.001 |
1.44 (1.32-1.57) |
< 0.001 |
|
eGFR categories (mL/min/1.73 m2) |
0.192 |
|
0.359 |
|
|
eGFR ≥ 90 |
1 |
1 |
|
|
|
60 ≤ eGFR < 90 |
1.25 (0.83-1.87) |
0.282 |
1.34 (0.73-2.43) |
0.344 |
|
45 ≤ eGFR < 60 |
1.21 (0.64-2.28) |
0.554 |
1.91 (0.83-4.40) |
0.127 |
|
30 ≤ eGFR < 45 |
0.995 (0.37-2.66) |
0.991 |
2.27 (0.71-7.26) |
0.166 |
|
eGFR < 30 |
4.26 (1.28-14.18) |
0.018 |
4.55 (0.39-52.79) |
0.226 |
3) Multivariable logistic regression analysis for clinical outcomes (Poor functional outcome at discharge) in patients with acute ischemic stroke in LAA and CE.
|
|
Poor functional outcome at discharge |
|||||
|
LAA |
CE |
|||||
|
n |
OR (95% CI) |
p-value |
n |
OR (95% CI) |
p-value |
|
|
eGFR categories (mL/min/1.73 m2) |
204 |
0.561 |
104 |
|
0.834 |
|
|
eGFR ≥ 90 |
61 |
1 |
22 |
1 |
|
|
|
60 ≤ eGFR < 90 (n=243) |
104 |
0.96 (0.51-1.79) |
0.893 |
57 |
1.17 (0.41-3.33) |
0.772 |
|
45 ≤ eGFR < 60 (n=67) |
22 |
0.56 (0.21-1.50) |
0.248 |
16 |
2.08 (0.48-9.0) |
0.327 |
|
30 ≤ eGFR < 45 (n=28) |
7 |
0.88 (0.25-3.17) |
0.847 |
7 |
0.95 (0.11-8.48) |
0.963 |
|
eGFR < 30 (n=21) |
10 |
2.19 (0.44-11.0) |
0.340 |
2 |
0.04 (0-1257334.67) |
0.709 |
Q: I would say that increase in NIHSS score of >2 points would be defined as neurologic deterioration.
A: Thank you very much for your valuable comments. We chose NIHSS ≥ 4 for the neurologic deterioration that was considered more conservative and used in other clinical studies as well. However, the definition of neurologic deterioration is variable among studies. We additionally performed statistical analysis for neurologic deterioration defining an NIHSS score of ≥ 2 points as you commented (listed below in Tables 1) and 2)). A statistically significant difference was not found among the eGFR groups, which was consistent with our results. We added the results for neurological deterioration (NIHSS ≥ 4) in tables 3 and 4.
1) Neurologic deterioration (NIHSS score of ≥ 2) in patients with acute ischemic stroke according to eGFR categories.
|
All patients (n = 1420) |
eGFR ≥ 90 mL/min/1.73 m2 (n = 429) |
60 mL/min/1.73 m2 ≤ eGFR < 90 mL/min/1.73 m2 (n = 731) |
45 mL/min/1.73 m2 ≤ eGFR < 60 mL/min/1.73 m2 (n = 164) |
30 mL/min/1.73 m2 ≤ eGFR < 45 mL/min/1.73 m2 (n = 63) |
eGFR < 30 mL/min/1.73 m2 (n = 33) |
p-value |
|
|
Neurologic deterioration |
127 (8.9) |
40 (2.8) |
58 (4.1) |
20 (1.4) |
4 (0.3) |
5 (0.4) |
0.576 |
2) Multivariable logistic regression analysis of the association between eGFR and neurologic deterioration in patients with acute ischemic stroke.
|
|
Neurologic Deterioration (NIHSS ≥2) |
Neurologic Deterioration (NIHSS ≥4) |
||
|
adjusted OR (95% CI) |
p-value |
adjusted OR (95% CI) |
p-value |
|
|
eGFR (mL/min/1.73 m2) |
|
0.331 |
|
0.931 |
|
eGFR ≥ 90 |
1 |
|
1 |
|
|
60 ≤ eGFR < 90 |
0.82 (0.54-1.25) |
0.363 |
0.90 (0.43-1.90) |
0.788 |
|
45 ≤ eGFR < 60 |
1.29 (0.73-2.28) |
0.391 |
0.94 (0.33-2.66) |
0.900 |
|
30 ≤ eGFR < 45 |
0.60 (0.57-4.29) |
0.355 |
1.20 (0.35-4.17) |
0.772 |
|
eGFR < 30 |
1.56 (0.08-0.18) |
0.392 |
0.43 (0.05-4.05) |
0.459 |

Round 2
Reviewer 2 Report
The authors sufficiently responded.